# Incorporation of Nanocatalysts for the Production of Bio-Oil from *Staphylea holocarpa* Wood

**DOI:** 10.3390/polym14204385

**Published:** 2022-10-17

**Authors:** Yiyang Li, Guanyan Li, Yafeng Yang, Xiangmeng Chen, Wanxi Peng, Hanyin Li

**Affiliations:** Henan Province Engineering Research Center for Forest Biomass Value-Added Products, School of Forestry, Henan Agricultural University, Zhengzhou 450002, China

**Keywords:** biomass, renewable energy, wood, catalyst, nanotechnology, pyrolysis

## Abstract

Biomass has been recognized as the most common source of renewable energy. In recent years, researchers have paved the way for a search for suitable biomass resources to replace traditional fossil fuel energy and provide high energy output. Although there are plenty of studies of biomass as good biomaterials, there is little detailed information about *Staphylea holocarpa* wood (*S. holocarpa*) as a potential bio-oil material. The purpose of this study is to explore the potential of *S. holocarpa* wood as a bio-oil. Nanocatalyst cobalt (II) oxide (Co_3_O_4_) and Nickel (II) oxide (NiO) were used to improve the production of bio-oil from *S. holocarpa* wood. The preparation of biofuels and the extraction of bioactive drugs were performed by the rapid gasification of nanocatalysts. The result indicated that the abundant chemical components detected in the *S. holocarpa* wood extract could be used in biomedicine, cosmetics, and biofuels, and have a broad industrial application prospect. In addition, nanocatalyst cobalt tetraoxide (Co_3_O_4_) could improve the catalytic cracking of *S. holocarpa* wood and generate more bioactive molecules at high temperature, which is conducive to the utilization and development of *S. holocarpa* wood as biomass. This is the first time that *S. holocarpa* wood was used in combination with nanocatalysts. In the future, nanocatalysts can be used to solve the problem of sustainable development of biological resources.

## 1. Introduction

One of the sustainable development goals of the United Nations 2030 agenda is to ensure access to affordable, reliable, and sustainable energy. Increasing the use of renewables in the global energy landscape is essential. Biomass is a substance formed by the metabolism of organisms that are usually derived from plants, industry, or agricultural and forestry wastes [1,2,3]. It can be a source of each energy treatment stage, especially for wood. Wood biomass is primarily intended for sustainable energy, fully integrating forestry and timber sectors [4,5,6]. Bioenergy-based production of biomass resources is gaining global attention, as it plays a critical role as an essential substitute for fossil energy [7,8,9]. Studies reported that biomass can serve as a good sustainable resource in the textile manufacturing field [10,11]. Biomass energy is currently being analyzed globally and formulate policy measures [12,13]. There is plenty of evidence that biomass wastes are suitable sources for bioenergy materials [14,15,16,17].

In many European countries, selected woody crops are used as raw materials for industrial and energy applications of biomass energy [18]. Forest biomass energy resources show great value and potential as important sustainable energy and renewable sources [19,20]. The use of bioenergy to develop renewable and clean biofuels can help to alleviate the worsening world energy crisis [21,22]. Crude bio-oil can also be produced in the process of thermochemical conversion of biomass and may contain different types of compounds, such as lignin-derived oligomers, alcohols, phenols, and aldehydes. So far, crude bio-oil has been used for wood fragrances, biodegradable polymers, resins, and fuel oil for stoves [23,24]. Chemical components in the extract can be used to extract bio-oil, which can be used as added energy. The raw material is a type of potential fuel and chemical raw material because of its availability and renewability. Dependence on oil and fossil fuels can be reduced in favor of the conducive to sustainable development energy [25,26,27].

Nanocatalysts have been widely explored in recent ten decades and are superseding conventional catalysts due to their high reactivity and selectivity [28,29]. Nanocatalysts have become one of the hot spots in the development of new functional materials at home and abroad, and they are also one of the most advanced technologies in the world [30]. Due to the size and unique characteristics of nanocatalysts (large surface area and high surface activity) and special properties (such as catalytic activity and stability), nanocatalyst has many properties that traditional catalysts cannot match [31]. The use of nanotechnology in bioenergy research has become a promising tool because of the advantages of low cost, high efficiency, and increased yield, and the use of nanocatalysts for biosynthesis has multiple advantages over other sources [32]. Recent advances in nanotechnology have been applied to environmental applications, energy, pharmaceuticals, and so on, and have gained a great deal of attention [33]. In addition, nanocatalysts play an important role in the production of biodiesel by increasing the reaction rate of the transesterification process and producing high-yield biodiesel [34]. Metal oxides such as Co_3_O_4_ and NiO have the ability to catalyze the conversion of petroleum into organic liquid products [35]. Nanocatalysts are one of the best solutions to current biomass utilization challenges in terms of their selectivity, energy efficiency, and time [36].

Although a large number of articles have reported studies on biomass, there is little information on the use of *Staphylea holocarpa* wood (*S. holocarpa*) as biomass energy in existing studies. Therefore, this study focused on *S. holocarpa* wood as the research object and focused on the potential of *S. holocarpa* wood as extracted bio-oil raw material. Fourier transform infrared (FTIR), gas chromatography mass (GC–MS), and liquid chromatograph quadrupole time of flight mass (LC–QTOF–MS) were used to study the characteristics and regularity of the extract, revealing the chemical components of the extract and the prospect of resource utilization. We also studied the role of nanocatalysts such as nickel (II) oxide (NiO), cobalt tetraoxide (Co_3_O_4_), and a combination of both catalysts (NiO+Co_3_O_4_) as well as exploring the effect of catalytic rapid pyrolysis during extract of bio-oil. Different nanocatalysts (NiO, Co_3_O_4_ and NiO+Co_3_O_4_) were added for further exploration and optimization of the pyrolysis process, and the thermal stability of *S. holocarpa* wood was characterized. The pyrolytic components of pyrolysis products of *S. holocarpa* wood were identified by using pyrolysis gas chromatography mass (PY/GC–MS), a thermal gravimetric analyzer (TGA), and thermal gravimetric analyzer–Fourier transform infrared (TGA–FTIR). In this study, various analytical methods were used to identify and analyze the active components of *S. holocarpa* wood extract and pyrolysed products in order to maximize bioenergy.

To our best knowledge, this is the first study that incorporates nanocatalysts in *S. holocarpa* wood and explores whether nanocatalysts can be used as potential additive to tackle the problem of future sustainable development of biomass energy. In addition, this study could provide fundamental insight on the comprehensive utilization and development of *S. holocarpa* wood, which could enhance the rapid development of the economy and maximize its utilization rate. The development and utilization of *S. holocarpa* wood would provide a scientific basis. In the future, the combination of wood and nanocatalysts will provide a new direction for biomass energy development, which will be conducive to solving the problem of sustainable development.

## 2. Materials and Methods

### 2.1. Sample Sources

The *S. holocarpa* samples were obtained from Xiaoqinling Nature Reserve, Lingbao City, China. They were dried and processed from fresh material into a powder. There were two experimental design settings in this study (Table 1). The first experimental design was extracting bioactive compounds with three different solvents (methanol, benzene/ethanol, and ethanol/methanol) from the wood–catalyst mixtures (Table 1a). Another experimental design was to set up a control group and three experimental groups. In the experimental group, wood powder samples of about 5 g were first mixed with different catalyst treatments as mentioned in Table 1b. The catalysts used in this study were 99.5% MACKLIN nano–NiO (30 nm) and 99.5% pure nano–Co_3_O_4_ (30 nm), and the amount of catalyst added in each sample was 0.05 g. Both nanocatalysts of nano–Co_3_O_4_ and nano–NiO were made by Shanghai Macklin biological Co., Ltd. in China.

### 2.2. Solvent Extractive

About 10 g of *S. holocarpa* wood powder was extracted with 20 mL of solvents, as mentioned in Table 1a. The pure extractive was obtained after a series of Soxhlet extraction processes as cited. The extractives were examined with FTIR, GC–MS analysis, and LC–QTOF–MS analysis.

FTIR analysis was performed using the Nicolet iS10 (Thermo Fisher, Waltham, MA, USA) instrument. A thin potassium bromide (KBr) disk was prepared from a mixture of KBr and the catalyst–wood samples at a ratio of 70:1 using mortar and pestle. The KBr disk was then loaded into an FTIR spectrophotometer at wavelengths from 400 cm^−1^ to 4000 cm^−1^ for 64 scans.

GS-MS analysis was analyzed with GC-MS (Agilent 7890B–5977A, Santa Clara, CA, USA). A HP-5MS column (30 mm × 250 μm × 0.25 μm) and an elastic quartz capillary column were used, along with a carrier gas of high-purity helium at a flow rate of 1 mL/min and a split ratio of 2:1. The temperature program for GC started at 50 °C and increased to 250 °C at a rate of 8 °C/min, followed by a further increase to 280 °C at a rate of 5 °C/min. The entire MS program scanned for a mass range of 30–600 amu with an ionization voltage of 70 eV and an ionization current of 150 μA. The ion source and quadrupole temperatures were set to 230 and 150 °C, respectively.

The ethanol/methanol extractives were analyzed by an Agilent 1260/1290 (USA) HPLC+6530/6545/6550 QTOF detector. The following LC and MS parameters were provided by Agilent Co.: The chromatographic column was Agilent Eclipse Plus C18 (2.1 mm × 100 mm, 1.8 μm). Positive ion modes in the mobile phase included 0.10% (*v*/*v*) formic acid (A) and with 0.10% (*v*/*v*) acetonitrile (B). One mM ammonium fluoride (or ammonium formate) (A) and acetonitrile (B) were in negative ion mode. The flow rate, column temperature and post column time were 0.30 mL/min, 40 °C and 5 min. Gradient elution [Time (min), B (%)] was [0, 5], [2, 5], [20, 100], [25, 100] in turn. MS detection mode was positive ion mode/negative ion mode. The source, drying gas flow (L/min), and drying gas temperature were AJS/ESI, 15 L/min (QTOF 6550)/7 L/min (QTOF 6530/6545), and 200 °C (QTOF 6550)/325 °C (QTOF 6530/6545), respectively. The nebulizer gas flow, sheath gas velocity, sheath gas temperature, capillary voltage, and scan mass range program were 35 psig, 11 L/min, 350 °C, 3.5 kV (Positive ion mode)/3.0 kV (Negative ion mode) and 50–1200 m/z, respectively. The reference ion was 121.0509 (64.0158), 922.0098 (Positive ion mode); and 112.9855 (68.9958), 1033.9881 (Negative ion mode).

### 2.3. The Physicochemical Properties of the Catalyst–Wood Mixture

The physicochemical properties of the catalyst–wood mixture (Table 1b) were examined with three main analyses: TGA, TGA–FTIR, and PY/GC-MS.

TGA analysis was performed using a thermogravimetric analyzer (TG–Q50, TA Instruments, New Castle, DE, USA). The analysis settings were as follows: nitrogen gas (N_2_) as a carrier gas with a 60 mL/min release rate. Temperature optimization was performed in four settings: 750 °C, 60 °C/min; 850 °C, 60 °C/min; 950 °C, 20 °C/min; and 950 °C, 100 °C/min. The TG and DTG curves were obtained and compared for thermal stability in order to study the rate of change of mass of *S. holocarpa* wood samples.

TGA–FTIR analysis was analyzed by a combined TGA–FTIR analyzer (TG–Q500, TGA Instruments, New Castle, DE, USA with FTIR–Thermo Scientific Nicolet iS10, USA). The analysis settings were as follows: nitrogen gas (N_2_) as a carrier gas with a 60 mL/min release rate; temperature increase from room temperature to 950 °C at 5 °C/min. Three-dimensional (3D) TG–FTIR spectra were generated for the study on the pyrolysis volatiles and structures.

PY/GC–MS analysis was conducted with a PY/GC–MS spectrometer (CDS 5000–Agilent 7890B–5977A). The sample was pyrolyzed at 950 °C with a heating rate of 20 °C/MS. The gas produced in the pyrolysis process was then injected in the GC-MS analyzer. The analysis settings for the GC–MS were as follows: TR–5MS column with a capillary size of 0.25 μm × 0.25 mm × 30 m at a 28–500 amu scanning range; shunt rate at 50 mL/min; split ratio at 50:1; temperature setting in two stages (increase rate of 5 °C/min from 40–120 °C and increase rate of 10 °C/min from 120–200 °C). Multiple components detected by the PY/GC-MS were categorized into four groups aldehydes and ketones, acids and esters, alcohols and ethers, and hydrocarbons for better understanding and analysis of the pyrolysis by products.

## 3. Results and Discussion

### 3.1. Extractive of S. holocarpa

#### 3.1.1. Analysis of FTIR

All spectra showed a typical lignin mode, and the characteristic absorption peaks of lignin were 1646, 1456, 1403, and 1323 cm^−1^ (Figure 1 and Table 2). The main structure of lignin was very similar to the characteristic band of the functional group. There was no significant change at 1273, 1081, 1046, or 878 cm^−1^, and *S. holocarpa* (methanol) was also weakened at 2980 cm^−1^ of cellulose, indicating that cellulose lignin was extracted [37]. The latter part was hydrolyzed [38]. Except for the different infrared absorption intensities all spectra were similar; the most typical band (1640 cm^−1^) represents the aromatic region of lignin [39,40]. In Figure 1, except for the peaks at 3446 and 1081 cm^−1^, the transmission intensities of all the peaks exceeded other values. As the carbon species changed, the transmission intensities of all peaks gradually decreased, indicating that these groups contained less carbon. There were mainly OH stretch at the peak of 3446 cm^−1^, –CH stretch at the peak of 2980 cm^−1^, C=C stretch at the peak of 1646 cm^−1^, CH stretch at the peak of 1456 cm^−1^, and CC stretching vibration at the peak of 1270 cm^−1^. The peak of 1046 cm^−1^ shows CO stretching. The 3742–2980, 2980–2863, and 1646–878 cm^−1^ ranges were the main concentrated parts of the absorption peaks of the three extract samples.

*S. holocarpa* wood’s methanol, benzene/ethanol and ethanol/methanol three extracts of light transmittance have changed in different degrees. The results showed that the main organic chemical components were ketones, acids, ethers, and so on [41]. In particular, the structure containing O and fat almost disappeared, which in turn showed that the intensity of the absorption peaks in the corresponding band weakened to varying degrees. The change in the functional group reflected the oxidation reaction activity of *S. holocarpa* wood to a certain extent. In addition, the characteristic absorption peak (1000–690 cm^−1^) indicated that ether, benzene, phenol, alcohol, and acid were partly extracted (Figure 1). From the FTIR results, it can be seen that all three samples contained mixtures with complex organic components and extremely high levels of oxygen. This included ether, carboxylic acids, aromatic rings, and other oxygen-containing organic substances (Table 2). These components are also the main chemical components of biomass.

#### 3.1.2. Analysis of GC–MS

Three ion chromatograms obtained from GC–MS analysis are shown in Figure 2. From the results of GC–MS analysis, about 18, 12, and 16 compounds were identified in *S. holocarpa* (methanol), *S. holocarpa* (benzene/ethanol), and *S. holocarpa* (ethanol/methanol), respectively (see Appendix A). The beneficial chemical components detected were maltol, n-hexadecanoic, (Z,Z)–9,12–octadecadienoic acid, 2–ethyl–1hexanol, dihydroxyacetone, clindamycin thymol, and d-mannose (Figure 2).

The active chemical composition of *S. holocarpa* (methanol) is (Z,Z)–9,12–Octadecadienoic acid, which had the highest abundance of density (Figure 2a). The (Z,Z)–9,12–Octadecadienoic acid was produced from linoleic acid via expression of fully recombinant *E. coli* cells from the diol synthase of *Aspergillus nidulans* [42,43]. Compound maltol is a commonly used food additive, and studies reported that maltol is preventative for liver oxidative damage caused by alcohol [44,45]. The ruthenium cyme complex derived from maltol has antitumor properties [46], and maltol can treat inflammatory–bowel–disease patients with iron-deficiency anemia [47].

Interestingly, compound n–hexadecanoic acid was detected in high abundance in all three solvent types (Figure 2a–c). It has anti-inflammatory properties and used to treat rheumatic symptoms [48] Adventitious roots cultured in vitro contained two valuable biologically active compounds: isosorbide and n–hexadecanoic acid [49]. Compound 1–Hexanol, 2–ethyl was detectable in all three solvent types and showed highest abundance in the *S. holocarpa* (benzene/ethanol) extractive (Figure 2b). It is widely used for the production of petroleum additives, plasticizers, and ore dressings, as well as printing and dyeing for paints and films [50,51,52].

There were some other medicinally important compounds detected in the *S. holocarpa* wood extract, such as dihydroxyacetone, clindamycin, thymol, trimethoorim–supamethoxazole, and D–mannose. Dihydroxyacetone can be used to synthesize medicines for the treatment of cardiovascular diseases [53]. It can also be used in cosmetics to prevent excessive evaporation of skin moisture and to protect against ultraviolet radiation [54,55]. Clindamycin is a lincomycin antibacterial drug [56]. Clindamycin and Trimethoprim–Sulfamethoxazole provide a significant increase in resistance to s–aureus infection in children [57]. Thymol has antibacterial and antifungal effects [58] as well as neuroprotective effects [59]. D–Mannose regulates T cells and inhibits immunopathology [60,61].

With detailed analysis of the detected compounds, it can be seen that *S. holocarpa* wood contains many healthy and beneficial chemical components. The beneficial compounds in the extract could be useful in a wide range of industrial applications, such as biomedical, cosmetic, and food products. This study could provide scientific foundation for the research and development of *S. holocarpa* wood.

#### 3.1.3. LC–QTOF–MS Analysis

Compounds were detected in the ethanol/methanol powder from the LC–QTOF–MS analysis (Appendix A). Among all detected compounds, four main compounds–celastrol, rhododendrin, isocryptotanshinone, and arbutin showed a significant role in pharmaceutical and medicinal applications (Table 3).

Isocryptotanshinone is a natural bioactive product with anti-cancer effects [70], it’s an effective STAT3 inhibitor induction lung cancer cell apoptosis and promotion of autophagy [62]. One of the active chemical composition is celastrol, which is also known as South Snake. This is a natural product with a variety of biological activities; it has a strong antioxidant effect and can treat obesity [63]. It can also inhibit the growth of gastric cancer cells and induce autophagy and apoptosis [64]. Furthermore, a synergistic effect on palmitic acid-induced cardiomyocyte apoptosis has been described [65]. Rhododendrin is an active anti-inflammatory compound activity. It is an ideal drug for the treatment of inflammatory skin diseases such as psoriasis [66]. Rhododendron synthetic derivatives have potent tyrosinase inhibitory activity [67]. Arbutin is an ideal whitening agent for whitening cosmetics. In cosmetics, it can effectively whiten and remove skin as well as gradually fade and remove skin freckles, chloasma, melanin, acne, and age spots. It is also highly safe and has no side effects such as irritation or sensitization. It has good compatibility with various components of cosmetics. Furthermore, it is stable under ultraviolet irradiation [68,69]. Compared to GC–MS, LC–QTOF–MS is a more in-depth data test. It detects more small organic molecules and makes the detected data more accurate.

### 3.2. Incorporation of Nanocatalyst with S. holocarpa Wood

#### 3.2.1. Analysis of TGA

TG and DTG curves at different optimization temperature settings for all four types of nanocatalyst treatments were obtained and are displayed in Figure 3. All four nanocatalyst treatments were about the same at different temperatures and at different rates. However, with an increasing heating rate, the temperature range of the pyrolysis reaction also increased. The starting and termination temperatures of each stage of the pyrolysis process moved slightly toward high temperatures. Therefore, the pyrolysis process had four stages. Stage I was from the initial temperature to about 120 °C, which is the water-evaporation stage. In this stage, water evaporation resulted in mass loss [71]. The temperature range of stage II was 120 to 265 °C, which was the transition stage of preheating. The loss of mass at stage II was relatively slow, which showed that the pyrolysis rate was steady. The mass loss mainly owed to the depolymerization and recombination of a small amount of high–content polymer in the sample [72]. Stage III was the volatilization analysis stage and the main stage of mass loss. The mass loss range of the four samples is 265–390 °C (Figure 3a,b), 265–360 °C (Figure 3c) and 265–380 °C (Figure 3d), respectively. The temperature increased in *S. holocarpa* wood samples in this temperature range. The cellulose and hemicellulose were rapidly cracked to form a large amount of volatile gas, which caused mass loss and caused the TGA curve to drop sharply [73]. At this stage, the volatile analysis yielded 80% to 90% of the mass loss for the entire temperature range. The peak of pyrolysis of cellulose appearing at 330 °C was at first due to high polymerization that formed oligosaccharides, which in turn decomposed and formed small molecules of gas and condensable volatiles of macromolecules. The pyrolysis temperature of hemicellulose was generally close to 200 °C, and the pyrolysis peak occurs between 265 °C–390 °C after pyrolysis [74].

The stage IV was the carbonization stage. When the temperature exceed 400 °C, the residue gradually decomposes into carbon or ash, and the mass loss is minimal. This stage was mainly the pyrolysis of lignin in comparison with the pyrolysis of cellulose and hemicellulose. In the process, the temperature range of the lignin pyrolysis was wide, generally occurring at 200–500 °C, and lignin pyrolysis generated more coke. The aromatic ring structure in the initial thermal cracking product decomposed and condensed after 500 °C to form a small molecular substance [75]. The sample decomposition temperatures in Figure 3 are 230, 255, 240 and 245 °C, respectively, which indicates that the thermal performance of the samples was almost unchanged.

The results showed that the DTG_max_ values for 750 °C, 850 °C, 950 °C (20 °C/min), and 950 °C (100 °C/min) were 373, 371, 372, and 360 °C, respectively (Figure 3). The DTG curve showed that when the temperature was about 115 °C (Figure 3a,b) and 105 °C (Figure 3c,d), the mass loss rate of *S. holocarpa* wood sample was one peak. This is because the cellulose, hemicellulose, and lignin composition of *S. holocarpa* wood samples treated with different nanocatalysts changed. Between 265 and 400 °C, mainly the cleavage of some sugars and phenols can be observed. The four samples increased gradually at the DTGmax value in the pyrolysis rates. This showed that the use of nanocatalysts could promote the catalytic cracking of sugars and phenols and that nano−Co_3_O_4_ had the best catalytic effect [76].

#### 3.2.2. TG−FTIR Analysis

Three-dimensional TG−FTIR spectra of all four nanocatalyst treatments are shown in Figure 4. The process of TG−FTIR analysis can be split into four stages. During stage I, the pyrolysis temperatures of *S. holocarpa* wood, *S. holocarpa* wood/NiO, *S. holocarpa* wood/Co_3_O_4_, and *S. holocarpa* wood/NiO+Co_3_O_4_ were 74, 79, 68, and 66 °C, respectively. A slight shoulder was shown in the wave number of 4000–3400 cm^−1^, indicating that the pyrolysis process of *S. holocarpa* wood corresponded. The volatilization of free water and the weight loss of the sample at this stage were small. During stage II, at 74–220 °C, 79–225 °C, 68–215 °C, and 66–218 °C, the overall curve was even, which proved that the pyrolysis rate was steady [77]. One type consisted of small-molecule gases such as CO_2_, CO, CH, and H_2_O. The other type consisted of typical tar components such as phenol, aldehyde, and acid [78].

During stage III, the precipitation content of several light gases increased significantly, cellulose and hemicellulose decomposed rapidly and produced a lot of volatile gases, *S. holocarpa* wood had the most severe pyrolysis reaction, and weight-loss rate was highest at 390 °C. The fourth stage was the carbonization stage after 400 °C; the weight loss was extremely low and the residue slowly decomposed into carbon and ash. Above 500 °C, the aromatic ring structure in the initial thermal cracking product decomposed and condensed to form a small molecular substance. At this stage, CO was the main precipitated gas, and further, a small amount of CO_2_ was precipitated; no other products were precipitated [79]. At the same time, other substances were precipitated during the pyrolysis process, and absorption peaks appeared in the infrared spectrum in the range of 2500–2250 and 1500–1250 cm^−1^ (Figure 4). This suggested that the precipitation of macromolecules such as aldehydes, hydrocarbons, carboxylic acids, and alcohols is due to C-O stretching vibration and C-C skeleton vibration in *S. holocarpa* wood [80]. By FTIR, many compounds were determined, and their characteristic absorbances at 4000–3400, 3050–2650, 2400–2240, and 2230–2000 cm^−1^ were identified, representing CH_4_, CO_2_, H_2_O, and CO [81]. The C=O tensile absorbance was between 1880 cm^−1^ and 1620 cm^−1^, which represents aldehydes, ketones, and acidic organic components. Between 1600 cm^−1^ and 400 cm^−1^, the spectral intensity of pyrolysis volatiles was the strongest, due to the characteristic absorption rate of some organic compounds [82].

The results show that the spectrum (a) reaction of *S. holocarpa* wood was relatively gentle and the spectrum (b–d) reaction with nanocatalysts was more intense, especially the spectrum (c) reaction (Figure 4). This showed that metal oxide catalysis was beneficial to the pyrolysis of *S. holocarpa* wood and that nano−Co_3_O_4_ had a better catalytic effect. The approximate maximum weight-loss temperature was at 340–390 °C, releasing a large number of complex organic volatiles, including carbonyl and oxy groups, which represented carboxyl, ketone, aldehyde, and alcohol. These are the main components of bio-oil. They can also synthesize a variety of drugs, pesticides, and their intermediates, and have a wide range of industrial application prospects. The results of TG−FTIR showed that *S. holocarpa* wood could be used to produce extracted bio-oil by pyrolysis. In addition, the size of metal nanoparticles had a great influence on catalytic function. Future TG−FTIR studies can start by changing the types of catalysts and reducing the particle sizes of the catalysts.

#### 3.2.3. Analysis of Py/GC–MS

*S. holocarpa* wood, *S. holocarpa* wood/NiO, *S. holocarpa* wood/Co_3_O_4_, and *S. holocarpa* wood/NiO+Co_3_O_4_ were identified with 124, 80, 90, and 100 active compounds, respectively, in the Py/GC–MS analysis (Appendix A). Each of the compounds detected was categorized into four main groups (Figure 5). *S. holocarpa* wood had 124 compounds: acid ester 28 (12.56%), ether alcohol 10 (27.88%), aldehyde ketone 12 (3.97%), and hydrocarbon 74 (55.59%). There were 80 compounds in *S. holocarpa* wood/NiO, represented by acid ester 12 (10.35%), ether alcohol 13 (25.56%), aldehyde ketone seven (7.64%), and hydrocarbon 48 (56.45%). *S. holocarpa* wood/Co_3_O_4_ featured 90 compounds: acid ester 21 (6.67%), ether alcohol 11 (38.3%), aldehyde 10 (6.15%), and hydrocarbon 48 (48.88%). *S. holocarpa* wood/NiO+Co_3_O_4_ featured 100 compounds: acid ester 17 (37.7%), ether alcohol 17 (9.04%), aldehyde ketone 5 (3.87%), and hydrocarbon 61 (49.39%).

In comparison of all the samples, the highest content of *S. holocarpa* wood/Co_3_O_4_ was 51.12% (acid ester, ether alcohol and aldehyde ketone). The results showed that nano−Co_3_O_4_ was the best catalyst for the reaction, which can improve the biomass composition and maximize the biomass output. As nano−Co_3_O_4_ is a catalyst with high catalytic activity, the pyrolysis rate and efficiency could be greatly improved [83,84]. In other reports, nano−Co_3_O_4_ could effectively improve redox reactions [85,86]. In addition, the shape of a Co_3_O_4_ nanocatalyst was more important than its size dependence [87,88]. The Py/GC−MS results showed that nano−NiO is not as effective as nano−Co_3_O_4_. However, nickel oxide nanocatalysts studied by Dawood et al. could effectively catalyze the conversion of *Brachychiton populneus* seed oil into biodiesel with high catalytic efficiency [89]. This may be greatly related to the concentration and shape of the catalysts used [90]. The nanocatalysts were introduced into the biomass pyrolysis process to realize the catalytic reaction of pyrolysis gas. Compared with pyrolysis, catalytic cracking could achieve directional catalytic conversion, promote reaction balance, and obtain more target products [91]. The use of low-cost and high-efficiency nanocatalysts is of great significance to the development of renewable energy [92]. The results of this study are conducive to the use of *S. holocarpa* wood as a biomass energy material and provide a scientific basis for the development and utilization of *S. holocarpa* wood.

## 4. Conclusions

In this study, *S**. holocarpa* wood was used as the research object to prepare bio-oil. *S. holocarpa* wood had been proven to have healthy and beneficial chemical components and be useful in biomedicine, food additives, and industrial applications. This study also provided fundamental insight on the role of nanocatalysts in the pyrolysis process, in which multiple volatile compounds were detected during the reaction: important components of extracted bio-oil (including ketones, acids, and alcohols). The results showed that the nanocatalysts affected the composition of pyrolysis products in Py/GC−MS analysis. The wood samples containing nano–Co_3_O_4_ tended to have good catalytic activity, which could efficaciously promote the catalytic cracking of *S. holocarpa* wood. This study, exploring the relationship between wood biomass, pyrolysis, and nanocatalysts, is conducive to the development and utilization of *S. holocarpa* wood, in addition to advocating for contribution and scientific knowledges in green science and green synthesis application. This is the first study that incorporated nanocatalysts in *S. holocarpa* wood. The nanocatalysts showed great potential for the production of value-added products in the pyrolysis results of *S. holocarpa* wood. However, the possibility of high production remains to be confirmed. The combination of nanocatalysts and pyrolysis is an innovative method to improve pyrolysis products. In the future, the catalytic effect of pyrolysis could be improved by optimizing the morphology, concentration, type, and various pyrolysis setting parameters of nanocatalysts, so as to maximize the catalytic effect of pyrolysis.

## Figures and Tables

**Figure 1 polymers-14-04385-f001:**
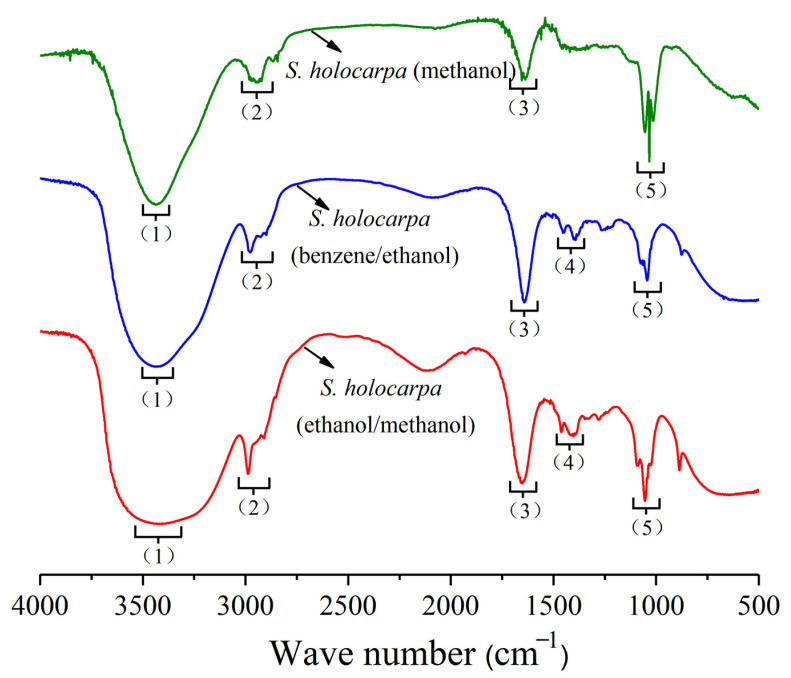
FTIR spectra of *S. holocarpa* (methanol), *S. holocarpa* (benzene/ethanol), and *S. holocarpa* (ethanol/methanol) samples.

**Figure 2 polymers-14-04385-f002:**
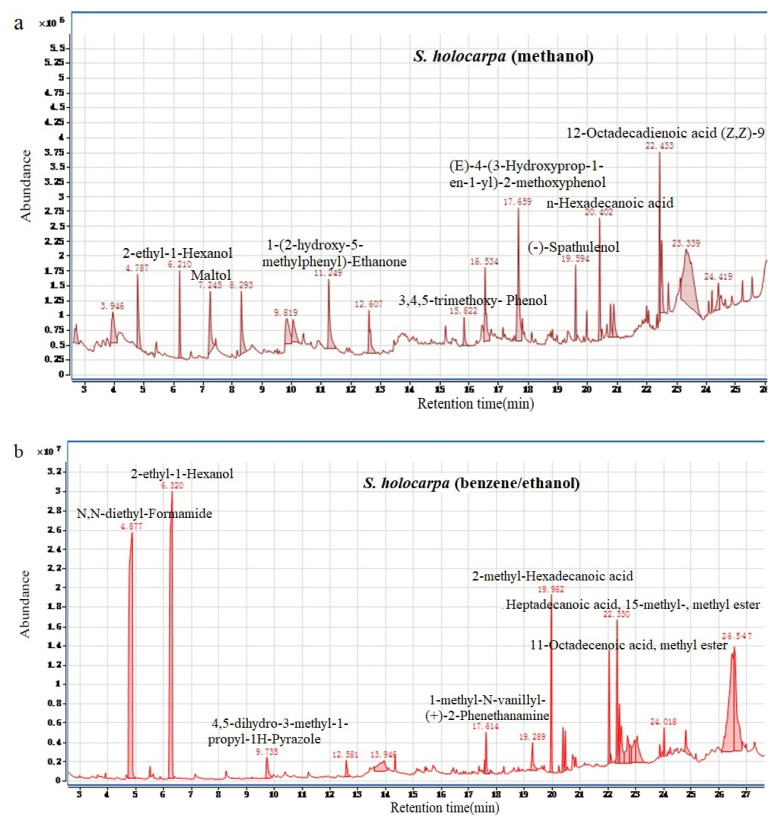
Total ion chromatogram of *S. holocarpa* (methanol) (**a**), *S. holocarpa* (benzene/ethanol) (**b**), and *S. holocarpa* (ethanol/methanol) samples (**c**).

**Figure 3 polymers-14-04385-f003:**
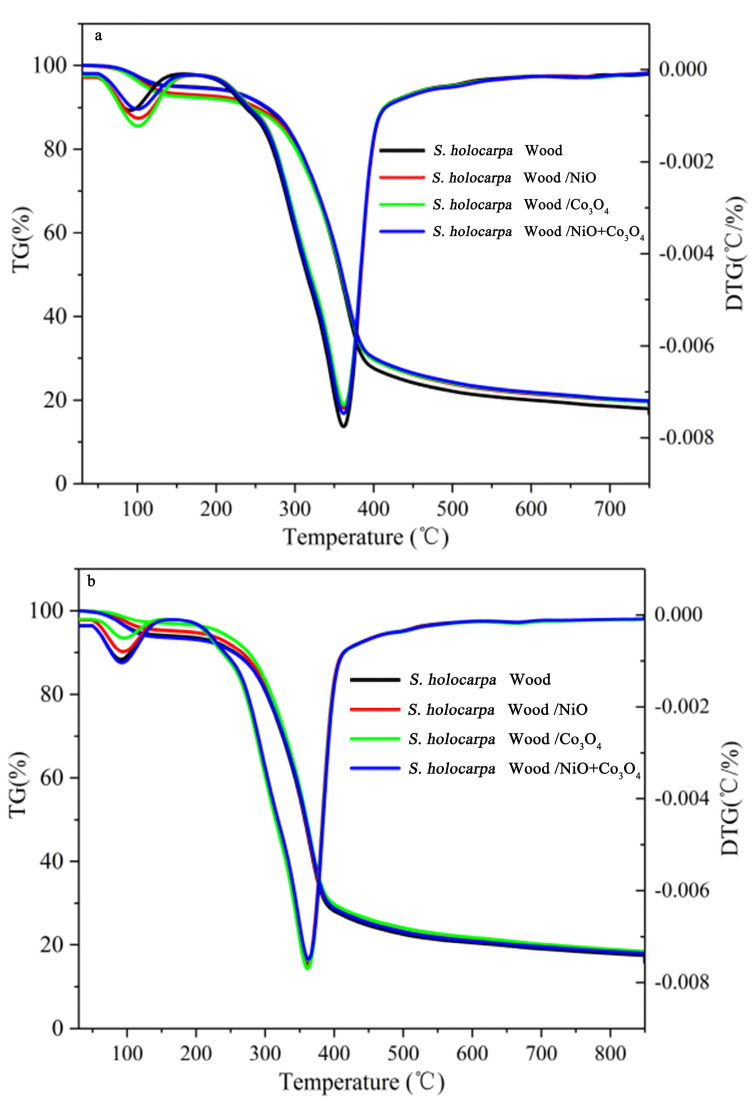
TG and DTG thermal curves samples of *S. holocarpa* wood with different catalytic treatment: (**a**) at a temperature of 750 °C (temperature rate of 60 °C/min), (**b**) at a temperature of 850 °C (temperature rate of 60 °C/min), (**c**) at a temperature of 950 °C (temperature rate of 20 °C/min), and (**d**) at a temperature of 950 °C (temperature rate of 100 °C/min).

**Figure 4 polymers-14-04385-f004:**
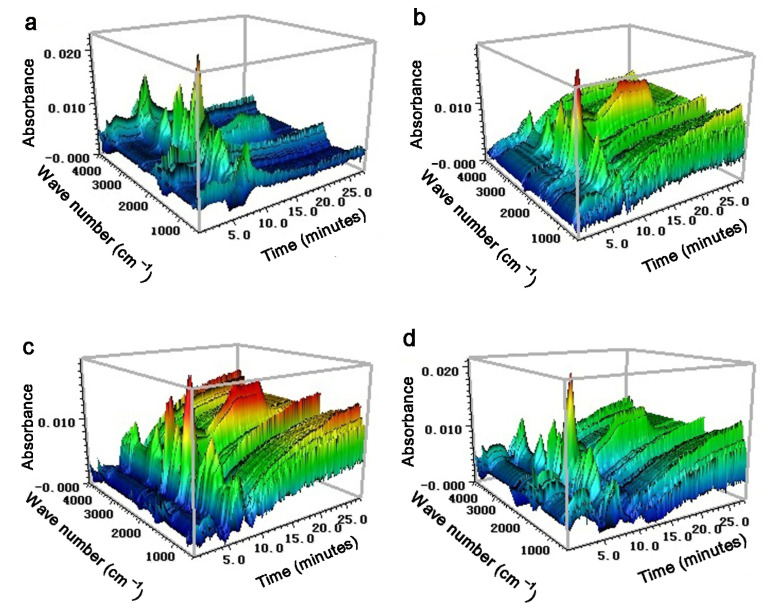
TGA and 3D FTIR spectrum of *S. holocarpa* wood (**a**), *S. holocarpa* wood/NiO (**b**), *S. holocarpa* wood/Co_3_O_4_ (**c**), and *S. holocarpa* Wood/NiO+Co_3_O_4_ (**d**).

**Figure 5 polymers-14-04385-f005:**
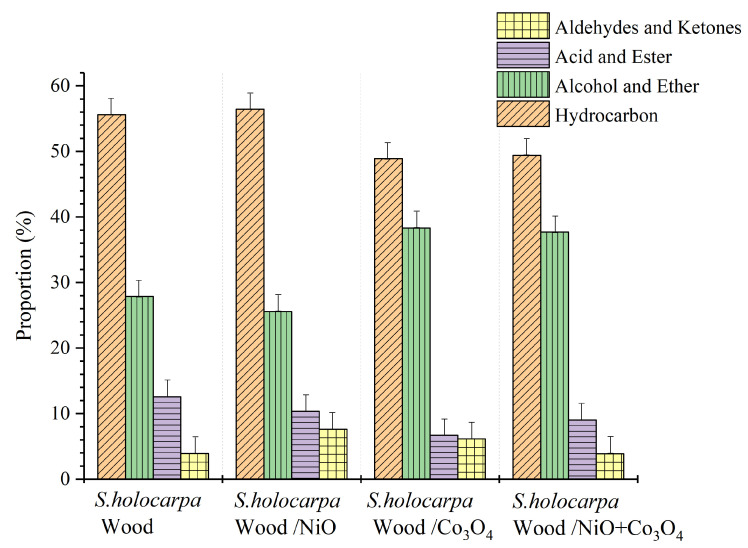
Proportion of pyrolysis products of *S. holocarpa* wood samples.

**Table 1 polymers-14-04385-t001:** Types of nanocatalyst treatments and extractive solvents.

**(a) Label**	**Extractive Solvents**
*S. holocarpa* (methanol)	Pure methanol
*S. holocarpa* (benzene/ethanol)	Mixture of benzene and ethanol in 1:1 ratio
*S. holocarpa* (ethanol/methanol)	Mixture of methanol and ethanol in 1:1 ratio
**(b) Label**	**Type of Nanocatalyst Treatments**
*S. holocarpa* Wood	*S. holocarpa* Wood (5 g)
*S. holocarpa* Wood/NiO	*S. holocarpa* Wood (5 g)+NiO (0.05 g)
*S. holocarpa* Wood/Co_3_O_4_	*S. holocarpa* Wood (5 g)+Co_3_O_4_ (0.05 g)
*S. holocarpa* Wood/NiO+Co_3_O_4_	*S. holocarpa* Wood (5 g)+NiO (0.025 g)+Co_3_O_4_ (0.025 g)

**Table 2 polymers-14-04385-t002:** The functional group and classification of compounds obtained from *S. holocarpa* wood extracts.

Frequency Range (cm^−1^)	Frequency (cm^−1^)			Functional Group	Classification of Compounds
	Methanol Extract	Benzene/Ethanol Extract	Methanol/Ethanol Extract		
(1) 3500–3000	3440	3444	3416	O–H stretching	Alcohol, carboxylic acids
(2) 3000–2800	2970	2976	2978	C–H stretching	Alkane
(3) 1680–1610	1641	1640	1644	C=C stretching	Alkenes
(4) 1470–1340	–	1454, 1391	1449, 1398	C–H bending	Alkanes
(5) 1200–1000	1053, 1034, 1016	1042	1081, 1044	C–O stretching	Alcohol, ether, carboxylic acids
(6) 900–690	–	874	878	C–H out of plane bending	Aromatic rings

**Table 3 polymers-14-04385-t003:** Four important medical compounds found in *S. holocarpa* wood.

No.	R_t_ (min)	Measured *m/z*	Identified Compound	Chemical Structure	Type of Compound	Molecular Formula	References
1	12.30	296.14	Isocryptotan–shinone	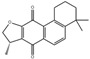	Ketones	C_19_H_20_O_3_	[62]
2	12.50	550.31	Celastrol	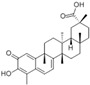	Alcohols	C_30_H_46_O_9_	[63,64,65]
3	13.10	328.15	Rhododendrin	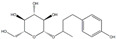	Organic compounds	C_16_H_24_O_7_	[66,67]
4	13.10	272.09	Arbutin	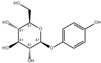	Glycosides	C_12_H_16_O_7_	[68,69]

## Data Availability

The data presented in this study are available on request from the corresponding author.

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
