# Peer review of "Incorporation of Nanocatalysts for the Production of Bio-Oil from Staphylea holocarpa Wood"

_polymers, 2022, doi:10.3390/polym14204385_

Round 1

Reviewer 1 Report

The research article submitted by Yang et al. as "Incorporation of Nano Catalyst for the Production of Bio-oil 2 from Staphylea holocarpa Wood" presents an extraction and characterization of bioactive compounds. For addressing such a topic, they use different characterization methods, such as TGA, TGA-FTIR, and PY/GC-MS, and the resulting work has interesting information. However, there are a few concerns about the present study, and I am suggesting publication in this journal after providing proper improvement in the revised version (major revision).

1-      The authors mention in the introduction section the that “there are plenty evidences that biomass wastes are great sources for bio-energy materials.” However, any reference was included. The section could be improved by including the suggested references: doi.org/10.1016/j.fuproc.2020.106427; doi.org/10.1007/s12649-020-01201-3; https://doi.org/10.3390/en14237977; doi.org/10.1590/1980-5373-MR-2022-0143

2-      Please, detail the methodology applied in the 2.1 and 2.2 sections.

3-      I do not find a real literature review in the discussion section and why authors have done this work, what is the scientific gap, and what is the novelty in this submitted research work.

4-       The discussion must be improved before publication. Could the authors provide a deeper/proper discussion section, by comparison to literature works, showing the relevance of the obtained results?

Author Response

Manuscript Number: polymers-1947226

Title: Incorporation of Nano Catalyst for the Production of Bio-oil from Staphylea holocarpa Wood

Dear respected editor and reviewers,

we would like to express our sincere gratitude to you for writing us the following constructive comments on our manuscript. Also, we appreciate very much for your willingness to check and help to improve the overall contents and quality of our manuscript with your precious time. Thank you so much for your comments and advice. We have made our best efforts to revise and improve our manuscript in an effort to acknowledge the reviewers’ comments accordingly. The comments from the reviewers are retyped below in italics, our responses are typed in normal black font, and the modifications done to the manuscript are also shown in red font. Thank you very much.

Comments from Reviewers

Reviewer #1:

Q1.  The authors mention in the introduction section the that “there are plenty evidences that biomass wastes are great sources for bio-energy materials.” However, any reference was included. The section could be improved by including the suggested references: doi.org/10.1016/j.fuproc.2020.106427; doi.org/10.1007/s12649-020-01201-3; https://doi.org/10.3390/en14237977; doi.org/10.1590/1980-5373-MR-2022-0143

Answer: We thank the reviewer very much for the comments. We have appended the reference to this sentence.

Action: “......There are plenty evidences that biomass wastes are great sources for bio-energy materials [14-17].”

Q2. Please, detail the methodology applied in the 2.1 and 2.2 sections.

Answer: We thank the reviewer very much for the comments. We detail the methodology applied in the 2.1 and 2.2 sections of the manuscript.

Action: “ 2.1. Sample sources

First experimental design was extract bioactive compound with three different(methanol, benzene/ethanol and ethanol/methanol)solvents from the wood-catalyst mixtures (Table 1a). Another experimental design is to set up control group and three experimental groups. In the experimental group, set on the powder wood samples about 5g were firstly mixed with different catalysts treatment as mentioned in Table 1b. The catalysts used in this study were 99.5% of MACKLIN nano-NiO (30 nm) and 99.5% of pure nano-Co3O4 (30 nm), and the amount of catalyst added in each sample was 0.05g.

2.2. Solvent extractive

GS-MS analysis was analyzed with GC-MS (Agilent7890B-5977A). A HP-5MS column (30 m × 250 μm × 0.25 μm) and an elastic quartz capillary column were used, with a carrier gas of high purity helium at a flow rate of 1 mL/min and a split ratio of 2:1. The temperature program for GC started at 50 oC and increased to 250 oC at a rate of 8 oC/min, followed by a further increase to 280 oC at a rate of 5 oC/min. The entire MS program scanned for a mass range of 30-600 amu with an ionization voltage of 70 eV and an ionization current of 150 μA. The ion source and quadrupole temperatures were set to 230 and 150 oC, respectively.

The ethanol/methanol extractives were analyzed by Agilent 1260/1290 HPLC + 6530/6545/6550 QTOF detector. The following LC and MS parameters were provided by Agilent Co. The chromatographic column was Agilent Eclipse Plus C18 (2.1 × 100 mm, 1.8 μm). 0.10% (v/v) formic acid (A), acetonitrile with 0.10% (v/v) formic acid (B) were positive ion modes in the mobile phase. 1mM ammonium fluoride (or ammonium formate) (A), acetonitrile (B) were negative ion mode. The flow rate, column temperature and post column time were 0.30 mL/min, 40 oC and 5min. Gradient elution: [Time (min), B (%)] was [0, 5], [2, 5], [20, 100], [25, 100] in turn. MS detection mode is positive ion mode/negative ion mode. The Ion source, drying gas flow (L/min) and drying gas temperature were AJS/ESI, 15 L/min (QTOF 6550)/ 7 L/min (QTOF 6530/6545) and 200 oC (QTOF 6550)/ 325 oC (QTOF 6530/6545). The nebulizer gas flow, sheath gas velocity, sheath gas temperature, capillary voltage and scan mass range program were 35 psig, 11 L/min, 350 oC, 3.5 kV (Positive ion mode)/3.0 kV (Negative ion mode) and 50-1200 m/z. Reference ion:121.0509(64.0158), 922.0098 (Positive ion mode); 112.9855 (68.9958), 1033.9881 (Negative ion mode).”

Q3.   I do not find a real literature review in the discussion section and why authors have done this work, what is the scientific gap, and what is the novelty in this submitted research work.

Answer: We thank the reviewer very much for the comments. We have revised the discussion section. The results of this work provide a scientific basis for the utilization and development of S. holocarpa wood as a biomass, and the use of low-cost and high-efficiency nanocatalyst can enhanced catalytic cracking and obtain more target products. This is the first study that incorporation of nanocatalyst in S. holocarpa wood. The results show that the nanocatalyst has great potential to produce value-added products in the pyrolysis results of S. holocarpa wood. But at the moment we are only experimenting in the laboratory and the possibility of high yield remains to be proven.

Q4. The discussion must be improved before publication. Could the authors provide a deeper/proper discussion section, by comparison to literature works, showing the relevance of the obtained results?

Answer: We thank the reviewer very much for the additional proposal. We have revised the discussion section and shown the relevance of the results obtained in this study by comparing them with other articles.

Action: “ By comparing all the , the highest content of S. holocarpab wood/Co3O4 was 51.12% (acid ester, ether alcohol and aldehyde ketone). The results show that nano-Co3O4 was the best catalyst in the log powder, which can increase the biomass composition and maximize the biomass output. As nano-Co3O4 is a catalyst with high catalytic activity, the pyrolysis rate and efficiency can be greatly improved [83, 84]. In other reports, nano-Co3O4 can effectively improve redox reactions [85, 86]. In addition, the morphology dependence of Co3O4 nanocatalyst is more important than its size dependence [87, 88]. The Py/GC-MS results show that nano-Nio is not as effective as nano-Co3O4. However, nickel oxide nanocatalyst studied by Dawood et al. can effectively catalyze the conversion of Brachychiton populneus seed oil into biodiesel with high catalytic efficiency [89]. This may be greatly related to the concentration and shape of the catalyst used [90]. The nanocatalysts were introduced into the biomass pyrolysis process to realize the catalytic reaction of pyrolysis gas. Compared with pyrolysis, catalytic cracking can achieve directional catalytic conversion, promote reaction balance, and obtain more target products [91]. The use of low-cost and high-efficiency nanocatalysts is of great significance for the development of renewable energy [92]. The results of this study are conducive to the use of S. holocarpa wood as a biomass energy material and provide a scientific basis for the development and utilization of S. holocarpa wood.”

Reviewer 2 Report

Graphical Abstract: Not provided. Suggestion to provide graphical abstract with 3D style so that it is more attractive. 

General comment:

This paper is study about the incorporation of nano-catalyst in S. holocarpa wood and exploring the nano-catalyst can be used as potential alternative material to tackle the problem of future environmentally sustainable development. Overall, this paper has well covered a matured technique. However, this paper needs to revise it carefully before it can be considered in high impact journal. Hope below comments will able to help to further improve the paper.

Specific comment:

Abstract:

-       An abstract is often presented separately from the article, so it must be able to stand alone. Hence the problem statement, aim, novelty and results of the study.

-       Please try to merge all information into a paragraph with some attractive and new findings.

Introduction:

-       Please mention the important of this study to society as well as industry.

-       Kindly refer some latest papers as it is highly relevant to this report. Example, Hydrogen production via sodium borohydride hydrolysis catalyzed by cobalt ferrite anchored nitrogen-and sulfur co-doped graphene hybrid nanocatalyst: Artificial neural network modeling approach

-       Revised Introduction section based on the structure below:

1st paragraph: Problem statement

2nd paragraph: Current ongoing solution

3rd paragraph: Proposed solution in this work.

4th paragraph: Summarized the current research novelty and objective of this work.

-       Problem statement of your introduction is not strong, need to discuss more about it.

Material and Method:

-       Authors need to specify all the source of materials were used

-       Unit should be written as (0.1 nm), a space should be included in between of the value and S.I. unit.

-       Suggest providing an additional figure to illustrate the process of the whole methodology.

Discussion:

-       The authors are encouraged to read this article for more scientific information, Current Trends and Future Prospects of Nanotechnology in Biofuel Production

-       The authors poorly explain their results in the discussion. Please find more information to support the facts. Some example can get from here: Synthesis of biodiesel from non-edible (Brachychiton populneus) oil in the presence of nickel oxide nanocatalyst: Parametric and optimisation studies

-       Kindly improve on the discussion. What is the significance of the results of the work?

-       There are some important information from this paper that authors are recommended to refer: Hybrid Pd50-Ru50/MXene (Ti3C2Tx) nanocatalyst for effective hydrogenation of CO2 to methanol toward climate change control

-       Please make sure the manuscript has been spelling and grammar checked before resubmission.

-        

Conclusions

-       Good but please include the limitations and what can be done for the future studies.

-       It is suggested to include additional information or clarifications to the methods and results sections to evaluate the manuscript's novelty and its significance to the field.

References

-       Kindly revise reference format according to the author guideline.

Additional materials for reading:

Catalytic hydrodeoxygenation of biomass-derived pyrolysis oil over alloyed bimetallic Ni3Fe nanocatalyst for high-grade biofuel production

Fabrication and Optimization of Nanocatalyst for Biodiesel Production: An Overview

Author Response

Manuscript Number: polymers-1947226

Title: Incorporation of Nano Catalyst for the Production of Bio-oil from Staphylea holocarpa Wood

Dear respected editor and reviewers,

we would like to express our sincere gratitude to you for writing us the following constructive comments on our manuscript. Also, we appreciate very much for your willingness to check and help to improve the overall contents and quality of our manuscript with your precious time. Thank you so much for your comments and advice. We have made our best efforts to revise and improve our manuscript in an effort to acknowledge the reviewers’ comments accordingly. The comments from the reviewers are retyped below in italics, our responses are typed in normal black font, and the modifications done to the manuscript are also shown in red font. Thank you very much.

Comments from Reviewers

Reviewer #2:

Graphical Abstract:

Q1. Not provided. Suggestion to provide graphical abstract with 3D style so that it is more attractive.

Answer: We thank the reviewer very much for the supplement. We have uploaded the graphical abstract to the system.

Action: “”

General comment:

This paper is study about the incorporation of nano-catalyst in S. holocarpa wood and exploring the nano-catalyst can be used as potential alternative material to tackle the problem of future environmentally sustainable development. Overall, this paper has well covered a matured technique. However, this paper needs to revise it carefully before it can be considered in high impact journal. Hope below comments will able to help to further improve the paper.

Answer: We thank you for your careful review and for given us a possibility to improve the quality of our manuscript. We revised the manuscript accordingly and detailed corrections are listed below point by point.

Specific comment:

Abstract:

Q1. An abstract is often presented separately from the article, so it must be able to stand alone. Hence the problem statement, aim, novelty and results of the study.

Please try to merge all information into a paragraph with some attractive and new findings.

Answer: We thank the reviewer very much for the comments. We have revised the abstract in the manuscript.

Action: “ ...... there is little detailed information about Staphylea holocarpa wood (S. holocarpa) as a potential bio-oil material. The purpose of this study is to explore the potential of S. Holocarpa wood as biological resources. Nano-catalyst cobalt (II) oxide (Co3O4) and Nickel (II) oxide (NiO) used to improve the production of  bio-oil from S. Holocarpa wood. The preparation of biofuels and the extraction of bioactive drugs were performed by the rapid gasification of nano-catalyst. The result indicated that the abundance chemical components had been detected in the S. holocarpa wood extract can be used in biomedicine, cosmetics and biofuels, and have a broad industrial application prospect. In addition, nano-catalyst cobalt tetraoxide (Co3O4) can improve the catalytic cracking of S. holocarpa wood and generate more bioactive molecules at high temperature, which is conducive to the utilization and development of S. holocarpa wood as biomass. This is the first time that S. holocarpa wood is used in combination with nano-catalyst. In the future, nano-catalyst can be used to solve the problem of biological resources sustainable development.”

Introduction:

Q1. Please mention the important of this study to society as well as industry.

Answer: We thank the reviewer very much for the supplement. In the last sentence of the introduction, we describe the importance of this research to society and industry.

Action: “......In the future, the combination of wood and nano-catalysts provides a new direction for biomass energy development, which is conducive to solving the problem of sustainable development. ”

Q2.  Kindly refer some latest papers as it is highly relevant to this report. Example, Hydrogen production via sodium borohydride hydrolysis catalyzed by cobalt ferrite anchored nitrogen-and sulfur co-doped graphene hybrid nanocatalyst: Artificial neural network modeling approach.

Answer: We thank the reviewer very much for the supplement. We have read carefully and add to the references.

Action: “......Recent advances in nanotechnology have been applied to environmental applications, energy, pharmaceuticals and so on and have gained a great deal of attention [33].”

Q3. Revised Introduction section based on the structure below:

1st paragraph: Problem statement

2nd paragraph: Current ongoing solution

3rd paragraph: Proposed solution in this work.

4th paragraph: Summarized the current research novelty and objective of this work.

Answer:  We thank the reviewer very much for the guidance about introduction .

Action: Since this change involves many paragraphs, please see our changes in the manuscript.

Q4. Problem statement of your introduction is not strong, need to discuss more about it.

Answer: We thank the reviewer very much for the comments. We have made changes in the manuscript.

Action: Please see our changes in the manuscript.

Material and Method:

Q1. Authors need to specify all the source of materials were used.

Answer: We thank the reviewer very much for the comments. We have added it to Section 2.1.

Action: “2.1. Sample sources 

.......Both, nanocatalysts of nano-Co3O4 and nano-NiO were made by Shanghai Macklin biological Co., Ltd. in China.”

Q2. Unit should be written as (0.1 nm), a space should be included in between of the value and S.I. unit.

Answer: We thank the reviewer very much for this question. We revised the unit in the text.

Action: “ ......The catalysts used in this study were 99.5% of MACKLIN nano-NiO (30 nm) and 99.5% of pure nano-Co3O4 (30 nm),...... ”

Q3. Suggest providing an additional figure to illustrate the process of the whole methodology.

Answer: We thank the reviewer very much for the comments. We have the whole methodology in the Graphical Abstract.

Discussion:

Q1. The authors are encouraged to read this article for more scientific information, Current Trends and Future Prospects of Nanotechnology in Biofuel Production.

Answer: We thank the reviewer very much for the comments. We have carefully read this article and add it as a reference.

Action: “......Nanocatalysts are one of the best solutions to current biomass utilization challenges in terms of their selectivity, energy efficiency and time [36].”

Q2. The authors poorly explain their results in the discussion. Please find more information to support the facts. Some example can get from here: Synthesis of biodiesel from non-edible (Brachychiton populneus) oil in the presence of nickel oxide nanocatalyst: Parametric and optimisation studies.

Answer: We thank the reviewer very much for the comments. We have refined the discussion. We read this article carefully and add it as a reference.

Action: “......The Py/GC-MS results show that nano-Nio is not as effective as nano-Co3O4. However, nickel oxide nanocatalyst studied by Dawood et al. can effectively catalyze the conversion of Brachychiton populneus seed oil into biodiesel with high catalytic efficiency [89]. ”

Q3. Kindly improve on the discussion. What is the significance of the results of the work?

Answer: We thank the reviewer very much for the comments. We have improved the discussion. The results of this work provide a scientific basis for the utilization and exploitation of S. holocarpa wood as biomass. The use of low-cost and high-efficiency nanocatalysts can intensify catalytic cracking and obtain more target products. It is of great significance to implement green chemistry, advocate green synthesis and develop renewable energy.

Action: “  By comparing all the , the highest content of S. holocarpab wood/Co3O4 was 51.12% (acid ester, ether alcohol and aldehyde ketone). The results show that nano-Co3O4 was the best catalyst in the log powder, which can increase the biomass composition and maximize the biomass output. As nano-Co3O4 is a catalyst with high catalytic activity, the pyrolysis rate and efficiency can be greatly improved [83, 84]. In other reports, nano-Co3O4 can effectively improve redox reactions [85, 86]. In addition, the morphology dependence of Co3O4 nanocatalyst is more important than its size dependence [87, 88]. The Py/GC-MS results show that nano-Nio is not as effective as nano-Co3O4. However, nickel oxide nanocatalyst studied by Dawood et al. can effectively catalyze the conversion of Brachychiton populneus seed oil into biodiesel with high catalytic efficiency [89]. This may be greatly related to the concentration and shape of the catalyst used [90]. The nanocatalysts were introduced into the biomass pyrolysis process to realize the catalytic reaction of pyrolysis gas. Compared with pyrolysis, catalytic cracking can achieve directional catalytic conversion, promote reaction balance, and obtain more target products [91]. The use of low-cost and high-efficiency nanocatalysts is of great significance for the development of renewable energy [92]. The results of this study are conducive to the use of S. holocarpa wood as a biomass energy material and provide a scientific basis for the development and utilization of S. holocarpa wood.”

Q4.  There are some important information from this paper that authors are recommended to refer: Hybrid Pd50-Ru50/MXene (Ti3C2Tx) nanocatalyst for effective hydrogenation of CO2 to methanol toward climate change control.

Answer: We thank the reviewer very much for the supplement. We have read the article carefully and add to the references.

Action: “ ......As nano-Co3O4 is a catalyst with high catalytic activity, the pyrolysis rate and efficiency can be greatly improved [83,84].   ”

Q5. Please make sure the manuscript has been spelling and grammar checked before resubmission.

Answer: We thank the reviewer very much for the comments. We make sure the manuscript has been checked for spelling and grammar.

Conclusions:

Q1.  Good but please include the limitations and what can be done for the future studies.

Answer: We thank the reviewer very much for the comments. We add the research limitations and future research directions at the end of the conclusion.

Action: “ In this study: S. holocarpa wood was used as the research object to prepare bio-oil.……The nanocatalyst showed great potential for the production of value-added products in the pyrolysis results of S. Holocarpa wood. However, the possibility of high production remains to be confirmed. The combination of nanocatalyst and pyrolysis is an innovative method to improve pyrolysis products. In the future, the catalytic effect of pyrolysis can be improved by optimizing the morphology, concentration, type and various pyrolysis setting parameters of nanocatellites, so as to maximize the catalytic effect of pyrolysis.”

Q2.   It is suggested to include additional information or clarifications to the methods and results sections to evaluate the manuscript's novelty and its significance to the field.

Answer: We thank the reviewer very much for the guidance. We have added additional information to the conclusion.

Action: Please see our changes in the manuscript.

References:

Q1.  Kindly revise reference format according to the author guideline.

Answer: We thank the reviewer very much for the comments. We have revised the reference format according to the author guideline.

Additional materials for reading:

Q1. Catalytic hydrodeoxygenation of biomass-derived pyrolysis oil over alloyed bimetallic Ni3Fe nanocatalyst for high-grade biofuel production

Fabrication and Optimization of Nanocatalyst for Biodiesel Production: An Overview

Answer: We thank the reviewer very much for the additional proposal. We have carefully read this article and add to the references.

Action: “ ......In other reports, nano-Co3O4 can effectively improve redox reactions [85, 86]. In addition, the morphology dependence of Co3O4 nanocatalyst is more important than its size dependence [87, 88]. ”

Once again, the authors are thankful to the Editor and Reviewers for providing us valuable feedback/suggestions on the manuscript to improve. We have thoroughly and carefully revised the relevant sections in the manuscript in accordance with the reviewers’suggestions. We hope that the reviewers will be satisfied with the updated version of the manuscript.

Best regards,

The authors

Round 2

Reviewer 1 Report

The authors have incorporated necessary changes to improve the manuscript. However, it still can be improved by addressing the following few points:

1-      In the introduction section, lines 43-47, the authors added information about bio-oil. However, it is a crude bio-oil which is different from the extracted bio-oil presented in this work. Thus, the authors could remove this sentence or add “Crude bio-oil” for this case (line 43) and “extracted bio-oil” in the remainder of the text.

2-      Line 176- Please add 1640 cm-1 instead of 164 cm-1

3-      Line 381- There is a missing word “By comparing all the ….(products?)

4-      Line 388- Please add NiO instead of Nio

5-      Line 417- I believe the word “nanocatellites” is a misprint of nanocatalysts.

6-      Line 418- Please add only a dot at the final of the sentence.

Author Response

Manuscript Number: polymers-1947226

Title: Incorporation of Nano Catalyst for the Production of Bio-oil from Staphylea holocarpa Wood

Dear respected editor and reviewers, we would like to express our sincere gratitude to you for writing us the following constructive comments on our manuscript. Also, we appreciate very much for your willingness to check and help to improve the overall contents and quality of our manuscript with your precious time. Thank you so much for your comments and advice. We have made our best efforts to revise and improve our manuscript in an effort to acknowledge the reviewers’ comments accordingly. The comments from the reviewers are retyped below in italics, our responses are typed in normal black font, and the modifications done to the manuscript are also shown in blue font. Thank you very much.

Comments from Reviewers

Reviewer #1:

The authors have incorporated necessary changes to improve the manuscript. However, it still can be improved by addressing the following few points:

Q1.    In the introduction section, lines 43-47, the authors added information about bio-oil. However, it is a crude bio-oil which is different from the extracted bio-oil presented in this work. Thus, the authors could remove this sentence or add “Crude bio-oil” for this case (line 43) and “extracted bio-oil” in the remainder of the text.

Answer: We thank the reviewer very much for the comments. We changed bio-oil into crude bio-oil in line 43-47, and changed bio-oil into extracted bio-oil in the main text of the manuscript.

Action: “......Crude bio-oil can also be produced in the process of thermochemical conversion of biomass and it may contain different types of compounds, such as lignin-derived oligomers, alcohols, phenols, and aldehydes. So far, crude bio-oil have been used as wood fragrances, biodegradable polymers, resins and fuel oil for stoves [23, 24].”

Q2. Line 176- Please add 1640 cm-1 instead of 164 cm-1.

Answer: We thank the reviewer very much for the comments. We have changed from 164 cm-1 to 1640 cm-1 in line 176.

Action: “......the most typical band (1640 cm-1) represents the aromatic region of lignin ”

Q3. Line 381- There is a missing word “By comparing all the ….(products?).

Answer: We thank the reviewer very much for the comments. We have modified it  on line 382.

Action: “.....By comparing all the samples......”

Q4. Line 388- Please add NiO instead of Nio.

Answer: We thank the reviewer very much for the proposal. We have changed from Nio to NiO in line 388.

Action: “The Py/GC-MS results show that nano-NiO is not as effective as nano-Co3O4. ”

Q5. Line 417- I believe the word “nanocatellites” is a misprint of nanocatalysts.

Answer: We thank the reviewer very much for the comments. We have changed from nanocatellites to nanocatalysts in line 418.

Action: “......type and various pyrolysis setting parameters of nanocatalysts, ......”

Q6. Line 418- Please add only a dot at the final of the sentence.

Answer: We thank the reviewer very much for the comments. We have made changes in line 419.

Action: “......so as to maximize the catalytic effect of pyrolysis. ”

Once again, the authors are thankful to the Editor and Reviewers for providing us valuable feedback/suggestions on the manuscript to improve. We have thoroughly and carefully revised the relevant sections in the manuscript in accordance with the reviewers’ suggestions. We hope that the reviewers will be satisfied with the updated version of the manuscript.

Best regards,

The authors

Reviewer 2 Report

Revisions have been made

Author Response

Manuscript Number: polymers-1947226

Title: Incorporation of Nano Catalyst for the Production of Bio-oil from Staphylea holocarpa Wood

Dear respected editor and reviewers, we would like to express our sincere gratitude to you for writing us the following constructive comments on our manuscript. Also, we appreciate very much for your willingness to check and help to improve the overall contents and quality of our manuscript with your precious time. Thank you so much for your comments and advice. We have made our best efforts to revise and improve our manuscript in an effort to acknowledge the reviewers’ comments accordingly. The comments from the reviewers are retyped below in italics, our responses are typed in normal black font, and the modifications done to the manuscript are also shown in blue font. Thank you very much.

Comments from Reviewers

Reviewer #2:

Revisions have been made

Answer: We thank you for your careful review and for given us a possibility to improve the quality of our manuscript.

Once again, the authors are thankful to the Editor and Reviewers for providing us valuable feedback/suggestions on the manuscript to improve. We have thoroughly and carefully revised the relevant sections in the manuscript in accordance with the reviewers’ suggestions. We hope that the reviewers will be satisfied with the updated version of the manuscript.

Best regards,

The authors
